# Diagnosis of Liver Fibrosis Using Artificial Intelligence: A Systematic Review

**DOI:** 10.3390/medicina59050992

**Published:** 2023-05-21

**Authors:** Stefan Lucian Popa, Abdulrahman Ismaiel, Ludovico Abenavoli, Alexandru Marius Padureanu, Miruna Oana Dita, Roxana Bolchis, Mihai Alexandru Munteanu, Vlad Dumitru Brata, Cristina Pop, Andrei Bosneag, Dinu Iuliu Dumitrascu, Maria Barsan, Liliana David

**Affiliations:** 12nd Medical Department, “Iuliu Hatieganu” University of Medicine and Pharmacy, 400000 Cluj-Napoca, Romania; popa.stefan@umfcluj.ro (S.L.P.);; 2Department of Health Sciences, University “Magna Graecia”, 88100 Catanzaro, Italy; l.abenavoli@unicz.it; 3Faculty of Medicine, “Iuliu Hatieganu” University of Medicine and Pharmacy, 400000 Cluj-Napoca, Romania; alexandru.padureanu@outlook.com (A.M.P.); miruna.dita@outlook.com (M.O.D.); bolchis.roxana@yahoo.com (R.B.);; 4Department of Medical Disciplines, Faculty of Medicine and Pharmacy, University of Oradea, 410087 Oradea, Romania; 5Department of Pharmacology, Physiology, and Pathophysiology, Faculty of Pharmacy, Iuliu Hatieganu University of Medicine and Pharmacy, 400347 Cluj-Napoca, Romania; 6Department of Anatomy, UMF “Iuliu Hatieganu” Cluj-Napoca, 400000 Cluj-Napoca, Romania; 7Department of Occupational Health, “Iuliu Hatieganu” University of Medicine and Pharmacy, 400000 Cluj-Napoca, Romania

**Keywords:** liver fibrosis, hepatic fibrosis, percutaneous liver biopsy, artificial intelligence, machine learning, computer scan, ultrasonography, digital pathology

## Abstract

*Background and Objectives***:** The development of liver fibrosis as a consequence of continuous inflammation represents a turning point in the evolution of chronic liver diseases. The recent developments of artificial intelligence (AI) applications show a high potential for improving the accuracy of diagnosis, involving large sets of clinical data. For this reason, the aim of this systematic review is to provide a comprehensive overview of current AI applications and analyze the accuracy of these systems to perform an automated diagnosis of liver fibrosis. *Materials and Methods***:** We searched PubMed, Cochrane Library, EMBASE, and WILEY databases using predefined keywords. Articles were screened for relevant publications about AI applications capable of diagnosing liver fibrosis. Exclusion criteria were animal studies, case reports, abstracts, letters to the editor, conference presentations, pediatric studies, studies written in languages other than English, and editorials. *Results***:** Our search identified a total of 24 articles analyzing the automated imagistic diagnosis of liver fibrosis, out of which six studies analyze liver ultrasound images, seven studies analyze computer tomography images, five studies analyze magnetic resonance images, and six studies analyze liver biopsies. The studies included in our systematic review showed that AI-assisted non-invasive techniques performed as accurately as human experts in detecting and staging liver fibrosis. Nevertheless, the findings of these studies need to be confirmed through clinical trials to be implemented into clinical practice. *Conclusions***:** The current systematic review provides a comprehensive analysis of the performance of AI systems in diagnosing liver fibrosis. Automatic diagnosis, staging, and risk stratification for liver fibrosis is currently possible considering the accuracy of the AI systems, which can overcome the limitations of non-invasive diagnosis methods.

## 1. Introduction

Chronic liver diseases (CLD) represent an important public health issue, accounting for significant morbidity and mortality globally and resulting in approximately 2 million deaths annually [1].

The precise etiology, geographic region, and presumably additional factors (sex, race, and socioeconomic status) have a significant impact on the incidence and prevalence of CLD [2].

Underlying etiology in CLD comprise alcohol-related liver disease, nonalcoholic fatty liver disease (NAFLD), chronic viral hepatitis B and C, autoimmune liver diseases (such as primary biliary cirrhosis, primary sclerosing cholangitis, and autoimmune hepatitis), hereditary diseases (Wilson’s disease, haemochromatosis, and alpha1-anti-trypsin deficiency) [3]. Regardless of the etiology, the course of CLD is characterized by a lengthy process of chronic parenchymal injury, prolonged inflammatory response, sustained activation of hepatic fibrogenesis, and continued activation of the wound healing response [4].

The development of hepatic fibrosis is a turning point in CLD, its presence and severity across the etiology being correlated with prognosis [3]. Liver fibrosis and fibrogenesis are key factors of the progression of any form of CLD towards liver cirrhosis and hepatic failure [4]. Liver fibrosis is characterized by hepatocellular damage (release of signals such as reactive oxygen species), the recruitment and activation of inflammatory cells (macrophages and lymphocytes generate multiple types of cytokines, including transforming growth factor-β and platelet-derived growth factor), and the excessive deposition of extracellular matrix proteins (differentiation of hepatic stellate cells towards myofibroblasts, dysregulated by cytokines) [5,6].

When fibrosis progresses, there is a worsening of the hepatic architecture, leading to bridging fibrosis and, eventually, cirrhosis (diffuse nodules of regenerating hepatocytes outlined by dense fibrotic tissue), causing hepatocellular dysfunction and distorted hepatic vasculature, which will result in hepatic insufficiency and portal hypertension [5].

Liver biopsy is the gold standard for fibrosis assessment because it allows detailed evaluation and localization and captures a larger amount of fibrosis [5]. However, its well-known drawbacks have made this procedure unappealing to doctors and patients (technical considerations, invasiveness, and potential severe complications) [7].

Considering this, efforts have been made in the last years for developing non-invasive strategies for assessing liver fibrosis. The several broad categories include serological markers (direct and indirect), imaging studies consisting of computed tomography (CT), magnetic resonance imaging (MRI), positron emission tomography–computed tomography (PET–CT), and methods assessing physical properties of the liver tissue (liver stiffness, attenuation, and viscosity) [2]. Methodologies that accurately and reproducibly evaluate liver anatomy and function without invasive procedures are urgently needed.

A new era of precision medicine in hepatology will begin once artificial intelligence’s (AI) ability to analyze data from digital imaging and pathology will be validated [8]. This will gradually revolutionize clinical practice, both from the perspective of understanding disease mechanisms and drug development. AI algorithms offer innovative prospects to forecast the likelihood of progression from early-stage CLDs toward cirrhosis-related consequences, with the goal of precision medicine [9]. For instance, certain AI programs have already been developed and have shown promising results regarding the screening of cirrhosis complications, such as esophageal varices and hepatocellular carcinoma [10,11,12]. Moreover, often requiring a thorough differential diagnosis and various imaging methods, focal liver lesions also represent a field in which AI could provide much needed assistance, with research suggesting an overall accuracy comparable with human experts [13]. State-of-the-art AI technologies are also being used in predicting the overall outcome of patients with liver tumors, as well as the overall response to therapy, by assessing the microvascular invasion before and after therapy [14,15]. Continuing initiatives must push past the tendency to oppose change and encourage the acceptance and use of these developing technologies.

In the last decade, AI applications used for automatic diagnosis have revolutionized radiology. AI algorithms can analyze images, such as X-rays, CT scans, and MRIs, to diagnose and classify abnormalities with a better precision than human experts. Furthermore, AI algorithms can recognize patterns and features that are not visible to human experts, making automatic diagnosis faster and more accurate. Because this technology can improve patient outcomes and reduce healthcare costs, the aim of this systematic review is to provide a comprehensive overview of current AI applications and analyze the accuracy of these systems in order to perform an automated diagnosis of liver fibrosis.

## 2. Materials and Methods

This systematic review was conducted in accordance with the preferred reporting items for systematic reviews (PRISMA) guidelines [16]. PubMed, EMBASE, Cochrane Library, and WILEY databases were searched for relevant publications about AI applications used for an autonomous diagnosis in liver fibrosis. The search terms included: (liver fibrosis OR hepatic fibrosis) AND (artificial intelligence OR machine learning OR neural networks OR deep learning OR automated diagnosis OR computer-aided diagnosis OR digital pathology OR automated ultrasound OR automated computer tomography OR automated magnetic imaging). We included articles indexed by the queried databases and returned by our search strategies, for which the full text was available, only in English, or if an English version was available. We considered all original research studies as eligible. Exclusion criteria were animal studies, case reports, abstracts, letters to the editor, conference presentations, pediatric studies, studies written in languages other than English, and editorials.

Two independent authors (S.L.P and A.I.) reviewed, for eligibility, titles, abstracts, and the full text of eligible articles. Data extraction was also conducted independently by both reviewers, with data on the authors’ names, year of publication, country or study population, sample size, study design, gender ratio, number and percentage of liver fibrosis patients, the method used to diagnose liver fibrosis, and artificial intelligence application being analyzed. Figure 1 shows the search strategy using the PRISMA flow diagram.

The initial search retrieved a total of 798 studies. We screened a total of 143 studies, and we excluded 119 articles as follows: irrelevant original studies to this review topic (n = 75), other languages (n = 16), conference abstracts (n = 5), articles not retrieved (15), and editorials or letters to the editor (n = 8). Finally, a total of 24 studies fulfilled our inclusion and exclusion criteria and were included in the systematic review as demonstrated in Figure 1.

## 3. Results

Histopathological analysis of liver tissue obtained via percutaneous biopsy is the current gold standard for identifying and staging hepatic fibrosis. However, there are some disadvantages accompanying biopsy, including peri-procedural pain, severe bleeding, and the potential of sampling bias due to the examination of only a limited area of liver parenchyma [17]. To overcome these drawbacks, non-invasive imaging-based approaches have been investigated as substitutes for biopsy: conventional MRI, magnetic resonance elastography (MRE), perfusion CT, and other experimental methods such as perfusion MRI, MR spectroscopy, and fibro CT [18].

Deep learning (DL) methods prove useful by aiding the clinician in making decisions. By combining the clinical point of view together with multiple paraclinical findings, such as laboratory and imaging findings, the diagnostic value rises. DL methods should be able to provide early identification of liver fibrosis, considering that early identification and accurate staging of liver fibrosis are critical for preventing or delaying clinical decompensation and the necessity for liver transplantation.

Clinically, it appears logical that in the case of severe liver fibrosis, the DL model focuses on both the liver and the spleen, because both organs undergo morphological changes when cirrhosis advances, as well as complications such as ascites, collateral circulation, and esophageal varices [19]. Therefore, these models should not only focus on the liver when describing liver fibrosis but also on the complications caused by advanced liver disease. These complications can be systemic, and for future perspectives, DL algorithms can be combined with blood parameters to help stage liver disease.

### 3.1. Artificial Intelligence Techniques and CT Imaging

The main studies analyzing the efficiency of AI algorithms in assessing liver fibrosis on CT images are illustrated in Table 1.

Yasaka et al. investigated if liver fibrosis could be effectively staged through deep learning techniques. They used a deep convolutional neural network (DCNN) trained and tested on 496 liver CT scans for the evaluation of the fibrosis stage in comparison to histopathological results. The study revealed that liver fibrosis could be staged with moderate performance based on dynamic contrast-enhanced portal phase CT images. For this particular AI model, the AUCs for diagnosing significant fibrosis, advanced fibrosis, and cirrhosis were 0.74, 0.76, and 0.73, respectively. Further improvements to the model are necessary in order for it to be used in clinical settings [20].

Li et al. conducted a study aimed at evaluating the performance of a residual neural network (ResNet) for staging liver fibrosis through plain CT images. The study involved liver CT scans from 347 patients with diagnosed CLD. Three different CT sections from adjacent levels were obtained for each patient, pre-processed through manual outlining of the interest area performed by two radiologists, and merged into a single sample for each patient. All the values obtained by the ResNet were the result of a cross-validation that was repeated five times between the CT image sample and the pathology report obtained from the assessment of liver biopsies. The accuracy of the ResNet model was higher than 0.82 for each category of fibrosis assessed through the METAVIR score, thus making the ResNet effective in evaluating fibrosis staging on plain CT images [21].

Using portal venous phase CT scans, Choi et al. created a deep learning system (DLS) to stage liver fibrosis. The DLS consists of two separate algorithms based on a convolutional neural network (CNN) in order to perform liver segmentation and fibrosis staging. In 707 of 891 individuals, the DLS correctly predicted the fibrosis stage, yielding a staging accuracy of 79.4%. The DLS created in this investigation was resilient across a variety of clinical settings and imaging situations with findings suggesting that the DLS’s accuracy in staging fibrosis was not reliant on CT scan methodology, patient demographic variables, or the presence of a liver focal mass. The diagnosis of intermediate stage fibrosis with the DLS was less accurate than the diagnosis of cirrhosis; the pathologic fibrosis stage was the only significant independent factor that significantly influenced the performance of the DLS [22].

Yin et al. used a new technique to better understand the interpretation of DL models when they staged liver fibrosis. The liver fibrosis staging network (LFS network) was created using contrast-enhanced CT scans taken during the portal venous phase of 252 individuals with histologically established liver fibrosis. Gradient-weighted Class Activation Mapping (Grad-cam) was used to locate where the LFS network focuses when predicting liver fibrosis stages. The corresponding location map revealed that the network strongly focused on the liver surface rather than the liver parenchyma when it came to a healthy liver, whereas in the case of cirrhosis (F4 liver fibrosis), the network focused more on the spleen and the central parts of the liver parenchyma [23]. The same group further used a combination of liver and splenic CT-based radiomics analysis to quantify liver fibrosis. Radiomics analysis, as opposed to DL, employs manually created features taken from CT scans. The model can show which types of symptoms on images are more essential to the model, and the results paralleled previous research. This means that the current radiomic analysis results might supplement the Grad-cam location maps by demonstrating the emphasis of DLS for predicting liver fibrosis stages [24].

Other directions for radiomics related studies include CT-texture analysis (CTTA) methods for the prediction of liver fibrosis and even differentiating between fibrosis grades. CTTA can quantify the heterogeneity and distribution of pixel or voxel grey levels on CT images. CTTA is based on extensive quantitative imaging characteristics that are undetectable to the naked eye and are created through numerous mathematical descriptors of the original picture. In their work, Budai et al. used CTTA software for processing liver CT images and predicting the fibrosis grade of each liver segment. A set of 354 CT images from 32 patients was used to extract quantitative parameters before texture analysis was performed. Results showed that CTTA-based models can not only detect fibrosis, but they also can differentiate between low- or high-grade fibrosis [25].

Wu et al. investigated the use of multi-slice spiral computed tomography (MSCT), which is centered on an AI segmentation algorithm, to diagnose liver cirrhosis and liver fibrosis. There were 112 patients included in the study and there were three indexes evaluated: hepatic arterial fraction (HAF), blood flow (BF), blood volume (BV), and mean transit time (MTT). Both patients with moderate liver fibrosis and those with substantial hepatic fibrosis had significantly higher HAF levels than those in the control group. Other indexes also achieved significant performance with authors concluding that larger sample sizes are needed to improve this method [26].

### 3.2. Artificial Intelligence Techniques and MRI Imaging

We found five studies assessing the accuracy of AI algorithms in diagnosing liver fibrosis on MRI images, as depicted in Table 2.

Nowak et al. conducted a study analyzing how a deep transfer learning (DTL) method can identify liver cirrhosis in standard transverse T2-weighted MRI images with accuracy compared to the assessments made by two radiologists. The study used two CNNs which were trained on a large natural data set of images obtained from the ImageNet archive. Then the transfer learning method was applied: the pre-trained CNN was adapted to identify liver cirrhosis in T2-weighted MRI scans. The AI was tested on 713 MRI scans from patients, 553 with confirmed liver cirrhosis and 160 with no history of liver disease. The DTL analysis utilized a single-slice MRI image, taken at the level of the caudate lobe for each entry. Two separate processing pipelines were used to analyze the images. The first one consisted of images priorly processed through a segmentation network and the second one utilized unsegmented images. The accuracy with which the DTL analysis correctly identified the presence of liver cirrhosis on the testing images was 0.97 for the pre-segmented set and 0.95 for the unsegmented set [27].

In the study conducted by Kato et al., the goal was to assess if the finite difference method paired with an artificial neural network (ANN) could be useful in identifying fibrosis in various acquisitions of MRI images. The study included 52 patients who underwent partial hepatectomy surgery for various liver tumors. The results obtained by the algorithm were compared to assessments made by two radiologists, and the fibrotic stage was also determined by a pathologist through semi-quantitative methods. On the samples, 10 areas of interest were marked by a radiologist prior to analysis. The ANN calculated seven texture parameters for each of the pre-determined areas on the samples and then compiled a probability for the presence of fibrosis in the whole liver. The AI model proved to be superior to the radiologists’ assessment, although no strong correlation between the radiologists’ grading and the ANN’s output could be established [28].

Hectors et al. created a DL algorithm based on gadoxetic acid-enhanced hepatobiliary phase (HBP) MRI in order to stage liver fibrosis. A secondary objective was to compare the diagnostic performance of DL vs. MRE. To reduce bias generated by the manual extraction of features and region of interest (ROI) placement as well as interobserver variability, it would be desired that DL models work fully automated. DL adopting CNNs can collect texture information in the initial convolutional layers, allowing picture texture analysis without the requirement for hand-crafted feature extraction. The group discovered that the algorithm performed well for predicting fibrosis severity with AUCs ranging from 0.77–0.91 for various fibrosis stages. Upon validation in different sets, the DL method may serve for noninvasive assessment of liver fibrosis without any need for extra MRI equipment, mainly because it had a similar performance compared to MRE [29].

Another MRI–DL technique combination which was recently introduced showed promising results in grading liver fibrosis after automatic segmentation of the liver. The method also uses a type of CNN for processing MRI Gadolinium ethoxybenzyl-diethylenetriaminepentaacetic acid (Gd-EOB-DTPA)-enhanced liver images from 121 livers pathologically confirmed as fibrotic or even cirrhotic (Ishak scores 0–6). It has been shown that CNNs with a U-shaped architecture are efficient at both segmenting organs and classifying them based on those segments. Because the model assigns an Ishak fibrosis score to each individual voxel, it is possible to make location-specific predictions about the amount of fibrosis. The approach functioned effectively, especially in situations where there was no fibrosis (Ishak 0) or cirrhosis (Ishak 6). Moderate fibrosis stages had a lower prediction rate, for which the authors suggest that the model’s capacity could be improved by integrating alternative sequences, such as T2 or diffusion-weighted imaging (DWI) [30].

Soufi et al. implemented a statistical shape modeling (SSM) technique based on partial least squares regression (PLSR), which directly uses the fibrosis stage as data to comprehend the liver shape and calculate a PSLR score. This was further used on the test data set to predict the fibrosis stage associated with this score in contrast-enhanced MR images. The SSM based on PLSR showed locally detailed variations in addition to generally recognized differences associated with liver fibrosis, such as shrinking of the entire right lobe or growth of the enlarged left lobe. The anterior section of the right lobe shrinks, while the caudate lobe and posterior part of the right lobe increase. As future perspectives, this method can be deeper explored by integrating the PLSR scores with other image features reflecting liver parenchyma properties, for example DL models combining CNNs as well as physiological information, such as serum or blood parameters, to increase fibrosis classification accuracy [31].

### 3.3. Artificial Intelligence Techniques and Ultrasonography

The main studies analyzing the accuracy of AI algorithms in detecting liver fibrosis on ultrasonography images are illustrated in Table 3.

The study conducted by Brattain et al. focused on developing an automated framework aimed to assess fibrosis grades in Sheer Wave Elastography (SWE) samples. The algorithm was meant to assess the quality of the SWE image, to automatically select an area of interest, and to decide whether that area presents a lesser or greater stage of fibrosis than stage F2. The study utilized several AI methods, and the best results were obtained by using the CNN model, with a performance assessed through the area under the curve of 0.89 [32].

Other imaging studies are also combined with machine learning (ML), as in, for example, the study conducted by Li et al. in which multiparametric ultrasound features served as input data for multiple ML algorithms. The types of parameters that were measured consisted of ultrasound images, radiofrequency data, and contrast-enhanced micro-flow images focused on a 2 cm ROI from the sixth liver segment. All these acquisitions, together with the ML models, are described as ultrasomics—a clinical decision support system based on large amounts of data which can predict liver fibrosis staging, necroinflammatory activity, and steatosis degree. The models combining morphological and hemodynamic characteristics performed better. This discovery indicates that using multiparametric ultrasomics from various pathophysiological procedures might improve the effectiveness of the clinical decision support system. The authors conclude that multicentric, whole-liver studies should be considered to increase the robustness of the multiparameter ultrasomics analysis [33].

Xie et al. used four network model structure schemes—AlexNet, VGG-16, VGG-19, and GoogLeNet—to find the most appropriate CNN model for ultrasound images of liver fibrosis analysis. Therefore, 640 samples in total from 780 individuals with cirrhosis and chronic hepatitis B were chosen for analysis. The GoogLeNet model was chosen as the best network model, because it performs recognition more accurately than other models. With a batch size of 32, a learning rate of 0.0005 as the parameter of the model, and a total of 10 iterations, the GoogLeNet model has the best classification and recognition effect in the analysis of ultrasound images of liver fibrosis and may eliminate the subjectivity of manual classification and increase the precision of assessing the severity of liver fibrosis, allowing for complete liver fibrosis prevention and therapy [34].

Zhang et al. looked to demonstrate, in their study, how an ANN may provide a duplex US-based non-invasive grading evaluation for hepatic fibrosis using data from 239 patients with different stages of liver fibrosis, with respect to cirrhosis. Five ultrasonographic measurements—the liver parenchymal, spleen thickness, hepatic vein waveform, hepatic artery pulsatile index (HAPI), and hepatic vein damping index (HVDI)—were chosen as the input neurons, because statistical analysis revealed a difference between the fibrosis group and the cirrhosis group in these five variables. This model can accurately identify liver cirrhosis when utilizing ultrasonography, according to certain predictive indices, including sensitivity, specificity, misdiagnosis rate (MR), and ROC curves for the ANN [35].

Using a total of 13,608 ultrasound scans from 3446 patients who had surgical resection, biopsy, or transient elastography, Lee et al. aimed to develop a CNN for METAVIR score prediction using B-mode ultrasound images. The AUC of the CNN was 0.866 for the classification of significant fibrosis (F2 or greater) in the test set, and for the classification of liver cirrhosis (F4), the algorithm achieved an AUC of 0.857. Most importantly, when utilizing US pictures to identify cirrhosis (F4), the CNN surpassed five radiologists. In the simulated US examination utilizing the test set, the CNN system had an AUC of 0.857, which was higher than that of each radiologist (AUC range, 0.656–0.816) [36].

Gatos et al., with the clinical data of 126 patients, used an algorithm based on ML and a stiffness value clustering to classify CLD using ultrasonic SWE imaging. Two radiologists’ clinical evaluations produced accuracy results of 75.3% and 76.6%, as well as sensitivity/specificity results of 72.2/80.1 and 73.8/81.3, respectively, proving that, in identifying healthy people from CLD patients, the proposed system performed better than all clinical and automated investigations and expert radiologists [37].

### 3.4. Artificial Intelligence Techniques and Liver Biopsy

Table 4 illustrates the main findings of studies analyzing the efficiency of AI algorithms in detecting liver fibrosis on liver biopsies.

Astbury et al. examined the effectiveness of a DL model with simple color space thresholding and human assessment in determining scar percentage in picrosirius red (PSR)-stained liver sections obtained from 20 cirrhotic explant livers. A quantitative evaluation of collagen or elastin throughout the entire region can be carried out using a color space threshold based on hue, saturation, and brightness (HSB). As opposed to HSB thresholding, computational approaches, particularly those based on AI, should allow the collection of data from liver biopsies while also minimizing the subjectivity inherent in the scoring process. Despite the issue seemingly favoring computational methods, there was significant residual inconsistency in the calculated scar percentage by the DL algorithm, and human observers consistently outperformed these methods. Because intra- and interlaboratory staining variation significantly reduces consistent PSR quantitative measurements using computer-aided methods and the section age may contribute to intra-laboratory variation if a standard timeframe between sectioning and staining is not respected, these findings suggest that quality control measures such as staining standardization and color adjustment will be necessary if AI-assisted scoring of stains is to be widely used [38].

Sarvestany et al. conducted a retrospective cohort study aimed to identify patients with liver fibrosis of any cause by using ML algorithms (MLAs). The study used 1703 liver biopsy specimens and associated demographic data and laboratory parameters provided by the Toronto Liver Clinic and McGill University Health Centre for testing the MLAs. The five validation sets comprised biopsies and data originating from the same health care facilities. Five standard MLAs as well as a combination of standard MLAs were used to differentiate between F0, F1, and F2 fibrosis stages regarded as one category and stages F3 and F4 considered as the other category. The ensemble of five MLAs proved superior to the other MLAs studied and also to other fibrosis detection methods that are not based on imaging techniques, such as APRI, FIB-4, or ENS, in identifying stages F3 and F4. The study claims that such MLAs could be used in the future for the screening of cirrhosis and advanced stage fibrosis [39].

The study conducted by Matalka et al. used an automated quantification system (AQS) to evaluate the degree of fibrosis in specimens of liver biopsy. The aim of the AQS was to identify the architecture of the fibrosis in tested samples through the recognition of textures and shapes that were representative of the fibrous expansion in the parenchyma. All images were pre-processed for clarity and brightness and segmented for better analysis of structural differences differentiating fibrosis stages. The AQS performed two different tasks: the first one being to differentiate between samples without fibrosis and fibrous samples of any stage and the second one to classify each fibrous sample to one of the six categories of the Ishak scoring system. The study included 260 samples, 50 without fibrosis and 210 with various Ishak stages of fibrosis, divided into a training and a testing set. The AQS differentiated non-fibrous samples from samples with varying degrees of fibrosis with an accuracy of 98.46%. Regarding the second stage of the AQS process, the accuracy for the testing lot was 94.69%. To further test the model, nine more samples were introduced in the algorithm, and the results obtained from the AQS were compared to those of two pathologists. The correlation between the AQS and the pathologists’ results were 0.9648 and 0.9125, respectively, after correcting the overlapping of the 5th and 6th Ishak stages in the ASQ analysis [40].

Qiu et al. developed a radiomics model in order to accurately stage liver fibrosis and detect early-stage cirrhosis, using a feature extraction technique from the DWI-MRI images of 369 patients from a single hospital. A biopsy with histopathology interpretation was used as the standard reference, with 108 patients presenting with liver fibrosis and early-stage cirrhosis and 146 with a healthy liver. Two radiologists performed volume of interest (VOI) extraction from these MRI images [35]. For maximal accuracy, the research team compared two analysis plans, of which the most proficient one achieved an AUC of 0.973 (95% CI 0.946–1.000) for the training dataset and an AUC of 0.948 (95% CI 0.903–0.993) for the independent testing dataset used for validation. At the time, the ML-assisted DWI-MRI diagnostic tool demonstrated utility in assessing liver fibrosis staging, with the goal of eventually replacing invasive biopsy for this purpose [41].

Wei et al. conducted a prospective study in which an ANN was constructed in order to isolate and predict biomarkers for fibrosis reversal in 141 treatment-naïve HBV patients with fibrosis S2/S3 staging between two treatment groups [42]. One consisted of 2 years of Entecavir therapy, and the other was Entecavir alternating with Entecavir combined with pegylated interferon (Peg-IFN). Patients included in the study were assessed using serum biomarkers every 6 months and liver biopsies at baseline and after 1.5 years post-treatment. The dataset was randomly divided into a training (80% patients) and testing set (20% of patients) and detected AST (aspartate aminotransferase), PLT (platelet count), WBC (white blood cell), CHE (cholinesterase), LSM (liver stiffness measurement), ALT (alanine aminotransferase), and gender as statistically significant parameters for liver fibrosis reverse prediction, using cross-sectional validation for the ANN’s performance. As a result, with a sensitivity and specificity of 83.1% and 85.2%, respectively, and an AUC of 0.809 in accurately classifying fibrosis with liver biopsy as the gold standard, these markers could constitute an accurate tool for predicting fibrosis reverse after antiviral therapy [42].

Wang et al. proposed a radiomics-based DL-algorithm for assessing liver fibrosis staging that was trained and validated with 1990 images from 398 patients of shear wave elastography and achieved an AUC of 0.97 for F4, 0.98 for ≥F3, and 0.85 for F2 [43]. Its performance was compared to that of conventional 2D-SWE and serum biomarkers (APRI model, using ASL, ALT, and FIB-4), using liver biopsy as a reference standard. The DL classifier performed better than 2D-SWE and biomarkers for all fibrosis types when more than one elastography image per patient was used as input, with the exception of F2 fibrosis, where the fibrosis heterogeneity is greater. There was no statistically significant difference between DLRE and 2D-SWE. The images were randomly, without overlap, divided into training (1330 images from 266 patients) and testing (660 images from 132 patients). The 2D-SWEs were manually cropped into an ROI, and that was used as the input layer of the DL. The DLRE’s accuracy, as expected, increased with the number of ROI input images in the training set, up to three images, with no significant improvement in the AUC between three and five images [43].

This DL classifier represented a diagnostic efficacy of fibrosis staging similar to the histopathological interpretation and performed significantly better than conventional 2D elastography and biomarkers. Another valuable feature was the DLRE’s diagnostic consistency when given data from various hospitals, suggesting the classifier’s robustness. However, testing other ethnic groups could bring different results [43].

## 4. Discussion

Most studies assessing computer-aided diagnostic tools for fibrosis detection and staging need a reference standard to compare their accuracy with, namely, biopsy with histopathological interpretation. Different types of ML-algorithms have been used for maximal diagnostic accuracy, such as DL (CNN-based classifiers), support vector machines (SVM), automated quantification systems, and random forest classifiers. In most cases, model overfitting of feature selection was avoided by using independent validation sets [20,39,40,42], and/or other methods, such as the RELIEFF algorithm, bootstrapping, and k-fold cross-validation [21,42]. However, some studies with low AUCs and an appropriate population size for ML-algorithm performance should consider these methods for validation.

The AI’s diagnostic performance was compared to radiologists’ interpretation performance and other non-invasive tests that represent current fibrosis staging guidelines, such as aspartate aminotransferase-to-platelet ratio index (APRI), Fibrosis-4 score (FIB-4), and alpha-fetoprotein (AFP) [39,43], as well as imaging techniques, such as 2D elastography [44] and MRE [30], demonstrating the AI’s diagnostic superiority. These comparisons are significant because, while AI-assisted tools may not be accurate enough to replace the gold standard, they may outperform other non-invasive alternatives.

Additionally, an inappropriate population study size could raise the error probability in the statistical analysis. Studies presenting such an issue would need a global database expansion [28,38,44] or merely regarding subgroups, such as additional data on cirrhotic patients [32]. Furthermore, while some studies used controls, other classifiers have been trained on unbalanced data with no control patients or in regard to cirrhosis and fibrosis patient distribution.

Different AI-assisted non-invasive techniques have achieved different diagnostic performances. While some studies showed high AUCs of 0.948 (95% CI 0.903–0.993) when using DWI-MRI images’ features when extracting features from SWE for maximal classification accuracy [41], others had a low AUC only ranging from 0.72 to 0.77 for the classification of fibrosis stages F0 vs. F1-4 and moderate performance and stages F0-1 vs. F2-4, F0-2 vs. F3-4, and F0-3 vs. 4. This shows the level of influence on diagnosis accuracy that different types of image techniques have, with elastography being shown to be more prone to disease heterogeneity errors [45]. However, elastography diagnostic accuracy can be raised with the use of SVM [46,47] and DL.

On the same note, a systematic review concluded that AI-assisted ultrasonography of NAFLD showed the highest diagnostic performance of all AI-assisted tools for NAFLD or NASH diagnosis or fibrosis detection [48]. It yielded a sensitivity and specificity of 0.97 (95% CI: 0.91–0.99) and 0.98 (95% CI: 0.89–1.00), respectively, an AUC of 0.98, and low heterogeneity. The next highest in terms of diagnostic performance was the AI-supported clinical diagnosis of NAFLD, with a sensitivity and specificity of 0.75 (95% CI: 0.66–0.82) and 0.82 (95% CI: 0.74–0.88), respectively, and an AUC of 0.85 with a slightly higher degree of heterogeneity. AI-supported clinical data sets performed comparably to conventional TE and slightly lower than MRI. Consequently, the information gathered on patient admission could be used as a screening method for at-risk patients for NAFLD. On the other hand, AI-assisted diagnostic tools for NASH diagnosis and fibrosis staging achieved a sensitivity of 80% (95% CI: 0.75–0.85) and a specificity of 0.69 (95% CI: 0.53–0.82).

This integration of clinical features (e.g., BMI, laboratory markers, gender, and comorbidities) along with the non-invasive procedures as input to the AI classifier with great diagnostic results has been successfully achieved in other studies [33,39]. Radiomics feature selection in combination with ML algorithms has been used, with ROI or VOI selection from 2D-SWE and DWI-MRI images made by experienced radiologists [24,41,43].

AI-based systems can help overcome the limitations of non-invasive methods by providing a more accurate and reliable diagnosis and staging of liver fibrosis. By combining the technology used in NAFLD and liver cirrhosis automatic diagnosis, researchers can develop AI-based systems that can accurately diagnose and stage liver fibrosis. Moreover, with the increasing availability of electronic health records, AI-based systems can be used to identify patients at high risk of developing liver fibrosis and provide timely interventions to prevent disease progression [49].

A timely and accurate diagnosis of liver fibrosis is essential for avoiding poor prognosis. However, liver biopsy, the current gold standard for diagnosis, is invasive and costly, with limited accuracy due to sampling error and intra- and interobserver agreement. Hence, the ability to assess fibrosis staging, steatosis, and inflammation with non-invasive techniques is crucial. Several studies have shown that ML algorithms can accurately diagnose fibrosis staging, with DL (CNN-based classifiers), SVM, and random forest classifiers achieving high accuracy. Although these AI-assisted tools may not replace liver biopsy, they can outperform other non-invasive alternatives, such as biomarkers and imaging techniques. AI-assisted non-invasive techniques have immense potential in accurately diagnosing liver fibrosis, allowing for timely risk factor modification and appropriate treatment. Researchers must expand the global database and validate the models using independent validation sets, additional data on controls, and increase the population study size to reduce the error probability in statistical analysis.

Due to the high prevalence of CLD, together with the lack of an adequate non-invasive diagnosis tests that would try to replace the liver biopsy, the subject of implementing AI algorithms into the diagnosis and management of liver fibrosis is of great importance. In this systematic review, the main imaging and diagnosis methods of liver fibrosis have been included, namely liver ultrasound, CT, MRI, and liver biopsy.

Nevertheless, the findings of the previously mentioned studies need to be confirmed through clinical trials. However, many studies had discrepancies regarding methodology, design, and outcomes. For this reason, international collaboration on AI systems can improve outcomes and provide a useful tool to human radiologists.

## 5. Conclusions

The current systematic review provides a comprehensive analysis of the performance of AI systems in diagnosing liver fibrosis. Automatic diagnosis, staging, and risk stratification for liver fibrosis is currently possible considering the accuracy, sensibility, and specificity of AI systems, which is comparable to human experts.

## Figures and Tables

**Figure 1 medicina-59-00992-f001:**
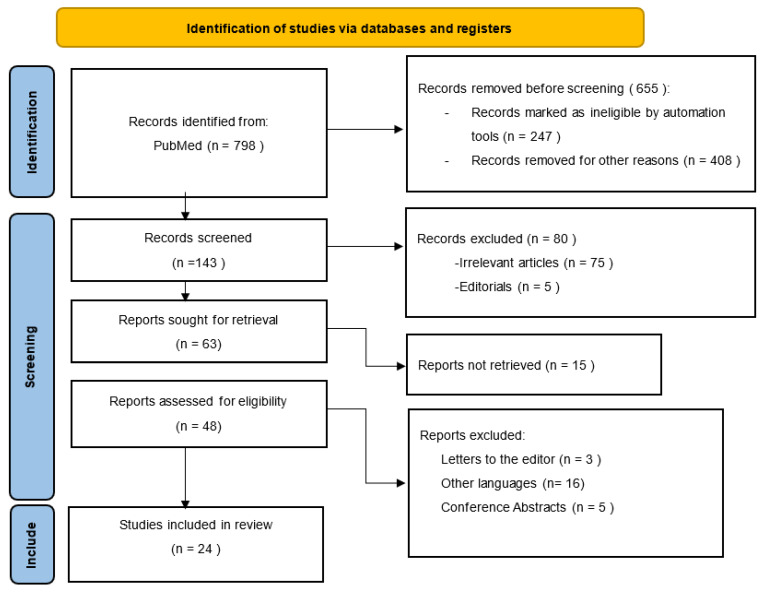
PRISMA flow diagram for study selection.

**Table 1 medicina-59-00992-t001:** Studies assessing AI techniques and CT imaging for the diagnosis of liver fibrosis.

First Author	Year	Total Number of Images	Diagnosis	Main Findings
Yasaka et al. [20]	2018	496	Liver fibrosis	Magnified CT images were analyzed by deep learning to diagnose and stage liver fibrosis, revealing a moderate correlation with histopathological staging.
Li et al. [21]	2020	1041	Liver fibrosis	The residual neural network (ResNet) is an efficient non-invasive diagnostic method for diagnosing liver fibrosis using plain CT images.
Choi et al. [22]	2018	7461	Liver fibrosis	The deep learning system was able to diagnose and stage live fibrosis with high accuracy (79.4%).
Yin et al. [23]	2021	252	Liver fibrosis	By using contrast-enhanced CT images and deep learning algorithms, liver fibrosis can be successfully diagnosed and staged.
Yin et al. [24]	2022	252	Liver fibrosis	Splenic radiomic features are an important and useful addition to hepatic radiomic features when staging liver fibrosis.
Budai et al. [25]	2020	354	Liver fibrosis	In order to differentiate between low- and high-grade fibrosis, CT texture analysis can be used for prognosis calculations of chronic liver disease.
Wu et al. [26]	2022	112	Liver cirrhosis and liver fibrosis	AI segmentation algorithms can be used to diagnose liver fibrosis in a clinical context.

CT: computed tomography.

**Table 2 medicina-59-00992-t002:** Studies assessing AI techniques and MRI imaging for the diagnosis of liver fibrosis.

First Author	Year	Total Number of Images	Diagnosis	Main Findings
Nowak et al. [27]	2021	713	Liver cirrhosis	Two pre-trained convolutional neural networks were successfully used to detect liver cirrhosis on standard T2-weighted MRIs.
Kato et al. [28]	2007	52	Liver fibrosis	The computer algorithm revealed a potential usefulness for the diagnosis of hepatic fibrosis.
Hectors et al. [29]	2021	355	Liver fibrosis	Deep learning algorithm, based on gadoxetic acid-enhanced MRI data, was comparable to MR elastography analysis.
Strotzer et al. [30]	2022	112	Liver cirrhosis and liver fibrosis	A multiphase Gd-EOB-DTPA-enhanced liver MRI was used to diagnose fibrosis stage or cirrhosis.
Soufi et al. [31]	2019	51	Liver fibrosis	PLSR-based SSM could help to better understand the variations associated with liver fibrosis staging and diagnosis.

MRI: Magnetic resonance imagine; MR: magnetic resonance; Gd-EOB-DTPA: Gadolinium ethoxybenzyl-diethylenetriaminepentaacetic acid; PLSR: partial least squares regression; SSM: statistical shape models.

**Table 3 medicina-59-00992-t003:** Studies assessing artificial intelligence techniques and ultrasonography for the diagnosis of liver fibrosis.

First Author	Year	Total Number of Images	Diagnosis	Main Findings
Brattain et al. [32]	2018	3392	Liver fibrosis	A new method of diagnosis for liver fibrosis that is based on a single image per decision compared to previous methods which used 10 images per decision.
Li et al. [33]	2019	144	Chronic hepatitis B	Machine-learning-based analysis of ultrasonography images can help stage liver fibrosis.
Xie et al. [34]	2022	640	Chronic hepatitis B and cirrhosis	The GoogLeNet model shows promising results in terms of recognition of lesions and diagnosis.
Zhang et al. [35]	2012	239	Liver fibrosis or cirrhosis	The ANN model presented high sensitivity and specificity for the non-invasive diagnosis of liver fibrosis.
Lee et al. [36]	2020	13,608	Liver fibrosis	Deep convolutional neural network accurately classified the ultrasonography images for cirrhosis diagnosis.
Gatos et al. [37]	2017	126	chronic liver disease	Color information quantification, from SWE images, by machine-learning can dissociate between chronic liver disease and healthy patients.

ANN: artificial neural network.

**Table 4 medicina-59-00992-t004:** Studies assessing artificial intelligence techniques and liver biopsy studies for the diagnosis of liver fibrosis.

First Author	Year	Total Number of Images	Diagnosis	Main Findings
Astbury et al. [38]	2021	20	Liver cirrhosis	Standardization between staining methods is still very important, as computational tools cannot yet normalize samples when performing analysis.
Sarvestany et al. [39]	2022	1703	Liver fibrosis	MLAs are able to help differentiate between patients with different prognoses concerning chronic liver disease.
Matalka et al. [40]	2006	260	Liver fibrosis	The automated quantification system differentiated between normal biopsies and samples with liver fibrosis, with an accuracy of 98.46%, and classified each sample with fibrosis according to the Ishak scoring system, with a precision of 94.69%.
Qiu et al. [41]	2020	369	Liver fibrosis	Radiomics analysis of liver images can accurately diagnose liver disease, resulting in a superior diagnosis tool compared to liver biopsy.
Wei et al. [42]	2019	141	Liver fibrosis	The multi-variable model developed can be useful for the evaluation of the clinical evolution of patients with chronic HBV-induced liver fibrosis.
Wang et al. [43]	2018	1990	Chronic hepatitis B	Deep learning Radiomics of elastography (DLRE) is useful for the non-invasive staging of liver fibrosis in patients infected with HBV.

MLAs: Machine learning algorithms; HBV: Hepatitis B virus.

## Data Availability

Not applicable.

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
