# Peer review of "Diagnosis of Liver Fibrosis Using Artificial Intelligence: A Systematic Review"

_medicina, 2023, doi:10.3390/medicina59050992_

Round 1

Reviewer 1 Report

The systematic review as proposed is heavy with informations

i would prefer to do the systemetic review for AI performance in liver fibrosis separately for MRI, scan, ultrasound and liver biopsy and compare in different liver diseases mainly NAFLD where the impact and the insidious presence of fibrosis is very important to detect with the best qccurqcy qnd the lowest morbidity

Didactic learning message is lacking

nice exhaustive and review work though

Author Response

Please see the attachment below.

Thank you for your collaboration and let us know what we could further improve about our manuscript.

Reviewer 2 Report

This is a very intrested paper that attempted to evaluate the role of AI in the diagnosis of LF.

The papers is very well written. 

Comments:

Many of the studies included have flaws on the methodology section. How this is calculated as an error from AI. This need to be added in the doc

Can the authors run a more advance statistical analysis and progress the paper to meta analysis? 

Author Response

Please see the attachment below.

Thank you for your collaboration and let us know what we could further improve about our manuscript

Reviewer 3 Report

This systematic review proposes a novel insight into the use of artificial intelligence as part of the current diagnostic tools for liver fibrosis. The manuscript is well structured, providing a global description and discussion of main findings from the studies included through the systematic literature search. Nonetheless, there are some aspects that authors should correct:

-     Abbreviations are completely inconsistent through the manuscript:

o   Please, define AI in the abstract.

o   Line 130, CT is not defined.

o   Line 81, CT, MRI and PET-CT are used, while they haven’t been defined previously.

o   Line 163, artificial intelligence should be replaced by AI.

o   Line 202, CT is already defined.

o   Lines 219 and 308, AI is already defined.

o   Lines 338 and 339: abbreviation should be in brackets, not the definition.

-   Abstract should be significantly improved. Materials and methods are too specific, it is not needed to define the terms used in the search strategy. In the conclusions, authors have described main findings, therefore, this would fit better in the results subsection of the abstract. In general terms, materials and methods should be smaller, while results and conclusions should be the most emphasized part of the abstract.

-    Inclusion criteria should be also described in the methodology.

-    Please, indicate which automation tools were used for screening.

-    Figure 1 should be corrected. There are arrows and boxes not matching.

-  In the last paragraph of the “Materials and Methods” section of the main manuscript, the search screening described does not match that in Figure 1. For example, authors screened 128 studies (line 118), but in Figure 1 a total of 143 studies are screened. Please, provide a correct description of the search performed, being in accordance with Figure 1.

- Tables are properly designed and summarized main parts of the study, however the last column is not clear. This last column of every table should be replaced by key points obtained from each study. This means that instead of providing a long description of main objectives and findings of each study included (that is already described along the results section), authors should be able to state point by point the most relevant findings of each study in a schematic way. This would increase the impact of these results on the reader and would avoid duplicated information between the tables and the text.

-  Line 329: authors refer to table x, but there is not such table attached. Please, clarify this issue (if authors are referring to a table from another article or to a table from this article).  

Author Response

(The authors gave the same response as above.)

Reviewer 4 Report

Overall, the review article is well written and well structured. In my opinion authors include most of the relevant studies. Though I fell they can add more about use of AI in liver diseases in introduction part.

Author Response

(The authors gave the same response as above.)
